 SHORT REPORT

# Adult-born granule cells modulate CA2 network activity during retrieval of developmental memories of the mother

**Blake J Laham, Isha R Gore†, Casey J Brown†, Elizabeth Gould\***

Princeton Neuroscience Institute, Princeton University, Princeton, United States

**Abstract** Adult-born granule cells (abGCs) project to the CA2 region of the hippocampus, but it remains unknown how this circuit affects behavioral function. Here, we show that abGC input to the CA2 of adult mice is involved in the retrieval of remote developmental memories of the mother. Ablation of abGCs impaired the ability to discriminate between a caregiving mother and a novel mother, and this ability returned after abGCs were regenerated. Chemogenetic inhibition of projections from abGCs to the CA2 also temporarily prevented the retrieval of remote mother memories. These findings were observed when abGCs were inhibited at 4–6 weeks old, but not when they were inhibited at 10–12 weeks old. We also found that abGCs are necessary for differentiating features of CA2 network activity, including theta-gamma coupling and sharp wave ripples, in response to novel versus familiar social stimuli. Taken together, these findings suggest that abGCs are necessary for neuronal oscillations associated with discriminating between social stimuli, thus enabling retrieval of remote developmental memories of the mother by their adult offspring.

## eLife assessment

This paper reports a **valuable** set of new results. The main result is that the projection from adult-born granule cells in the dentate gyrus to the hippocampal subfield CA2 is necessary for the retrieval of a social memory formed during development, and **solid** evidence is provided to support this conclusion.

**\*For correspondence:**
goulde@princeton.edu

†These authors contributed equally to this work

**Competing interest:** The authors declare that no competing interests exist.

## Introduction

The earliest social memories arise as infants learn to recognize their first caregiver - most commonly, the mother. In humans, these early associations can be further strengthened by emotional valence and shared experiences as the child develops, forming memories of the mother that can last a lifetime (*Fivush, 2011*; *Schaal et al., 2020*). Mice can also recognize their mothers and distinguish them from unfamiliar mothers during the first postnatal week (*Laham et al., 2021*). These memories last well into adulthood, with a reversal in social preference at the time of weaning (*Laham et al., 2021*); pup offspring prefer investigating their own mothers over novel mothers, while adult offspring prefer investigating unfamiliar mothers over their own. In adulthood, the ability to discriminate between the caregiving mother and a novel mother persists despite any contact with the mother for over a month after weaning. Lower investigation times of the mother compared to a novel mother suggest retrieval of a remote social memory by adult offspring.

The CA2 region of the hippocampus is required for the ability to recognize caregiving mothers and distinguish them from novel mothers both during development and in adulthood (*Laham et al., 2021*). The CA2 is also necessary for short-term memory of peers during development and in adulthood (*Hitti and Siegelbaum, 2014*; *Smith et al., 2016*; *Laham et al., 2021*; *Diethorn and Gould, 2023*).

A growing literature suggests that rhythmic firing in the CA2 supports the social memory functions of this region. Phase-amplitude coupling (PAC) of low frequency theta rhythm with higher frequency gamma rhythm in the CA2 and the generation of sharp wave ripples (SWRs), high frequency bursts in the CA2, are increased by exposure to familiar peers and short-term consolidation of social memory for peers, respectively (*Zhu et al., 2023*; *Oliva et al., 2020*). In other hippocampal subregions, namely the CA1 and CA3, PAC and SWRs have been associated with novelty detection, memory consolidation, and memory retrieval of nonsocial stimuli and events over relatively short time frames (*Colgin, 2015*; *Fernández-Ruiz et al., 2019*; *Joo and Frank, 2018*; *Meier et al., 2020*; *Vivekananda et al., 2021*). These studies suggest that certain patterns of oscillatory firing may facilitate replay and pre-play of neuronal ensembles activated during specific experiences (*Joo and Frank, 2018*). However, no previous studies have examined whether PAC and SWRs are associated with retrieval of remote developmental memories.

The CA2 has been called a social memory 'hub' because it receives and integrates inputs from multiple brain areas (*Hitti and Siegelbaum, 2014*; *Oliva et al., 2020*; *Diethorn and Gould, 2023*), including from granule cells of the dentate gyrus (*Kohara et al., 2014*), a population known to undergo adult neurogenesis (*Song et al., 2012*). abGCs are important for discriminating between novel and familiar peers (*Pereira-Caixeta et al., 2018*; *Cope et al., 2020*), and although these cells are known to project to the CA2 (*Llorens-Martín et al., 2015*), the function of this circuit remains unexplored. Furthermore, although the dentate gyrus is known to influence hippocampal gamma oscillations, PAC, and SWRs in general (*Bott et al., 2016*; *Meier et al., 2020*; *Fernández-Ruiz et al., 2021*), no studies have examined the influence of abGCs on these oscillatory patterns in the CA2 specifically. Here, we investigated whether abGCs support the retrieval of developmental memories of the mother, and whether these cells contribute to socially relevant CA2 network activity.

## Results
### abGCs project to inhibitory interneurons and pyramidal cells of the CA2 region

In the CA2, we verified abGC projections using 3R-Tau, a marker of abGC axons (*Llorens-Martín et al., 2015*), as well as mature granule cell projections using ZnT3, a marker of mature mossy fibers (*Wenzel et al., 1997*; *Figure 1a and ai–aiii*). We further determined that abGC projections to the CA2 are preferentially associated with the proximal dendrites of parvalbumin-positive (PV +) interneurons labeled with cre-dependent AAV mCherry compared to the proximal dendrites of pyramidal cells immunolabeled with PCP4 (*Figure 1b–d*). Although we cannot determine whether a similar relationship exists with projections to distal dendrites or whether these connections represent active synapses, this preferential association with PV + interneurons over pyramidal cells mirrors abGC projection patterns to the CA3 (*Restivo et al., 2015*). Because the CA2 region is involved in the retrieval of remote developmental memories (*Laham et al., 2021*) and abGCs have been linked to the storage and retrieval of recently encountered mice (*Cope et al., 2020*), we next investigated whether abGCs are necessary for the retrieval of developmental memories of the mother.

### abGCs are necessary for retrieval, but not maintenance, of remote developmental memories

Using the GFAP-TK transgenic mouse model in which abGCs are ablated upon valganciclovir (VGCV) administration (*Snyder et al., 2011*), we found that adult GFAP-TK mice and CD1 littermates with control levels of abGCs investigate their own mother less than a novel mother (*Figure 1e–g*, *Figure 1—figure supplement 1a*), similar to what has been observed in C57 mice (*Laham et al., 2021*). Because adult mice prefer novel stimuli over familiar stimuli, lower investigation times of previously encountered mice compared to novel mice can be interpreted as recognition. Thus, our social investigation data in adult mice with intact abGCs suggest that these mice can recognize mothers as familiar even after being separated since weaning (at P21). In contrast, mice with depleted abGCs are no longer able to discriminate between their own mother and a novel mother, as indicated by investigation times (*Figure 1—figure supplement 1a*). No changes were observed in overall locomotion during social trials after abGC depletion (*Figure 1—figure supplement 2a*), suggesting that any physical activity differences between groups are not responsible for differences in social investigation

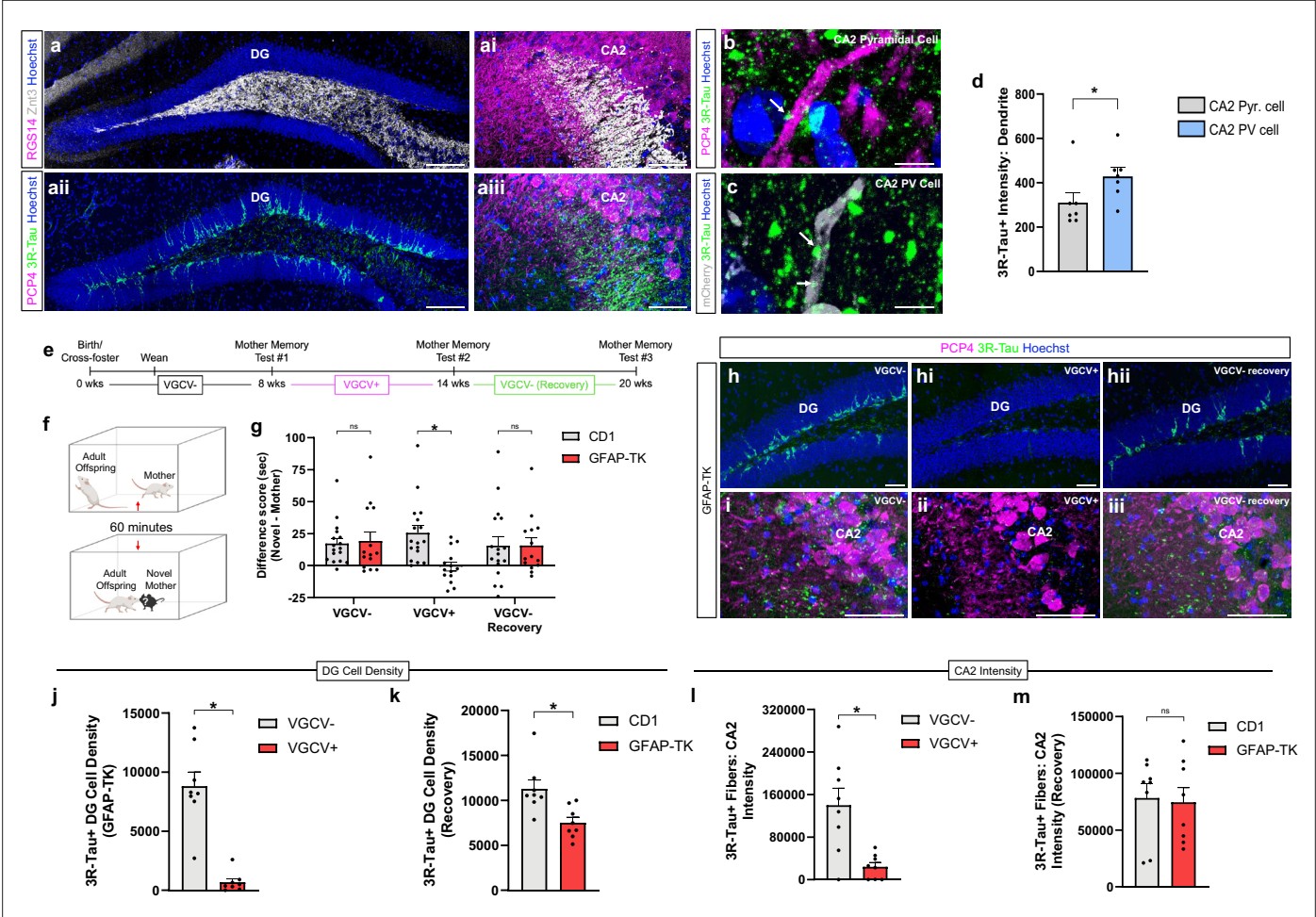

**Figure 1.** Adult-born granule cells support retrieval of developmental remote memories of the mother. (**a-aiii**) Confocal images of dentate gyrus granule cells, both mature (ZnT3+) and adult-born (3R-Tau+), and their robust projections to CA2. Confocal images of adult-born granule cell (abGC) 3R-Tau+axons (arrows) (**b**) near proximal dendrite of a CA2 PCP4 + pyramidal cell and (**c**) near proximal dendrite of a CA2 mCherry + PV+ interneuron. (**d**) abGC fibers (3R-Tau+) in the CA2 are more abundant near proximal dendrites (n=7; paired t-test: $t_6$=3.163, p=0.0195) of PV + interneurons (mCherry+) than those of pyramidal cells (PCP4+). (**e**) Timeline for behavioral experiment. (**f**) Schematic demonstrating direct social interaction assay used at three experimental time points. (**g**) Ablation of abGCs abolishes difference in investigation time between mother and novel mother (CD1: n=17; TK: n=14; two-way RM ANOVA: Genotype x Drug: $F_{2,58}$ = 5.352, p=0.0074; Šídák's test p=0.0021). After removing valganciclovir (VGCV) from rodent chow and allowing adult neurogenesis to recover over the course of 6 weeks, mice were able to discriminate between mother and novel mother again (p>0.999). (**h-hii**) Confocal images of abGCs in the DG at three different drug time points. VGCV administration produces a dramatic loss of (**h-hii**) abGCs (3R-Tau + cells) in DG (VGCV-: n=8; VGCV+: n=8; unpaired t-test: $t_{14}$=6.588, p<0.0001) and (**i-iii**) 3R-Tau+ fibers in CA2 (VGCV: n=8; VGCV+: n=8; unpaired t-test: $t_{14}$=3.488, p=0.0036). 6 weeks after removing VGCV from the rodent chow, DG abGC number (**j, k**) undergoes a 70% recovery (CD1: n=8; TK: n=8; unpaired t-test: $t_{14}$=3.218, p=0.0062), and (**l, m**) 3R-Tau+ fibers present in CA2 undergo a complete recovery (CD1: n=8; TK: n=8; unpaired t-test: $t_{14}$=0.7271, p=0.4791). *p<0.05, bars represent mean + SEM. Scale bars = 200 µm for **a, h, i**; 2 µm for **b, c**. ns = not significant.

The online version of this article includes the following source data and figure supplement(s) for figure 1:

**Source data 1.** Adult-born granule cells support retrieval of developmental remote memories of the mother.

**Figure supplement 1.** Total investigation times during social interaction.

**Figure supplement 1—source data 1.** Total investigation times during social interaction.

**Figure supplement 2.** Valganciclovir (VGCV) or clozapine-N-oxide (CNO) administration does not alter locomotion.

**Figure supplement 2—source data 1.** Valganciclovir (VGCV) or clozapine-N-oxide (CNO) administration does not alter locomotion.

times. Cessation of VGCV treatment in GFAP-TK mice enabled a 70% recovery of abGCs in the DG (*Figure 1h, j and k*) and complete recovery of abGC 3R-Tau+ axon intensity in the CA2 (*Figure 1i, l and m*). Coinciding with the recovery of adult neurogenesis, GFAP-TK animals regained the ability to discriminate between their mother and a novel mother with differential investigation times (*Figure 1— figure supplement 1a*). These results reveal that abGCs are not necessary for maintaining developmental social memories, but instead support retrieval of remote memories of the mother. Because it is known that abGCs project to the CA2 (*Llorens-Martín et al., 2015*), a region which has been linked to retrieval of developmental social memories (*Laham et al., 2021*), we next inhibited projections from abGCs to the CA2 to investigate how this circuit affects mother memory retrieval.

## abGC projections to the CA2 play a time-limited role in the retrieval of remote developmental memories

To determine if developmental social memory retrieval requires activation of the abGC-CA2 circuit, we used a tamoxifen-inducible double transgenic mouse where Gi-DREADD expression was confined to abGCs in a temporally specific way (Nestin-cre:Gi). Nestin-cre littermates were used as single transgenic controls. Targeted clozapine-N-oxide (CNO) infusion through bilateral cannula aimed at the CA2 enabled inhibition of Gi-DREADD + abGC axon terminals in the vicinity of the cannula in Nestin-cre:Gi mice. Previous studies have shown that abGCs are important for nonsocial memories only during a limited time after they are generated, i.e., 4–6 weeks after mitosis (*Gu et al., 2012*; *Song et al., 2012*; *Kropff et al., 2015*). Accordingly, tamoxifen-treated adult mice underwent behavioral testing at two post-injection time points (4–6 weeks and 10–12 weeks post-injection) to assess whether different aged abGCs are involved in developmental social memory retrieval (*Figure 2a–d*). As expected, we found that Nestin-cre:Gi mice investigated the mother less than the novel mother when vehicle was infused; however, inhibition of this pathway after CNO infusion in CA2 prevented this ability (*Figure 2e*, *Figure 1—figure supplement 1b*). There were no differences in locomotion at baseline or during the social trials in any groups, suggesting that the effects of social investigation time were not the result of physical activity differences (*Figure 2*, *Figure 1—figure supplement 2b and c*). CNO treatment in the same abGC cohort 6 weeks later (10–12 week-old abGCs) had no effect on Nestin-cre:Gi animals' ability to discriminate between their mother and a novel mother (*Figure 2f*, *Figure 1—figure supplement 1c*). These findings suggest that abGCs are required for retrieving remote social memories of the mother when the cells are immature (4–6 weeks post-mitosis), but not when they are mature (10–12 weeks post-mitosis), presumably because a younger abGC cohort has taken over this function. It should be noted that although cannula were placed over the CA2 region, we cannot rule out the possibility that CNO affected abGC terminals in the nearby CA3a region. An additional experiment explored if abGCs contribute to peer social memory consolidation (*Figure 2g*). Immature abGCs were silenced immediately after the investigation of a novel adult mouse. When reintroduced to the now familiar adult mouse 6 hr later, a time at which studies have shown that the effects of CNO on other behaviors have worn off (*Alexander et al., 2009*; *Whissell et al., 2016*; *Ray et al., 2011*), mice exhibited control-like social discrimination (*Figure 2h*), suggesting that abGC activity may not be required for social memory consolidation.

Our findings suggest that abGC projections to the CA2 play a role in retrieval of remote memories of the mother. However, to date, no studies have investigated how abGC inputs affect CA2 network activity. Because neuronal oscillations, including SWRs and PAC, in the CA2 have been linked to social recognition of recently encountered conspecifics (*Oliva et al., 2020*; *Zhu et al., 2023*), we next investigated whether abGC inputs to the CA2 modulate these events during exposure to mothers and novel mothers, specifically.

## abGCs are necessary for differential network activity in the CA2 during exposure to novel versus familiar social stimuli

To test whether abGCs affect CA2 SWRs, we recorded neuronal oscillations in the CA2 with and without inhibition of 4–6-week-old abGCs in separate groups of Nestin-cre and Nestin-cre:Gi mice (*Figure 3a and b*). We found that SWR production (*Figure 3c and d*) is increased during exposure to a social stimulus (*Figure 3e and k*), and that more SWRs are produced during novel mother exposure, presumably during encoding social memories, than during mother exposure, presumably during retrieval of developmental social memories (*Figure 3—figure supplement 1a and b*; *Figure 3—figure*

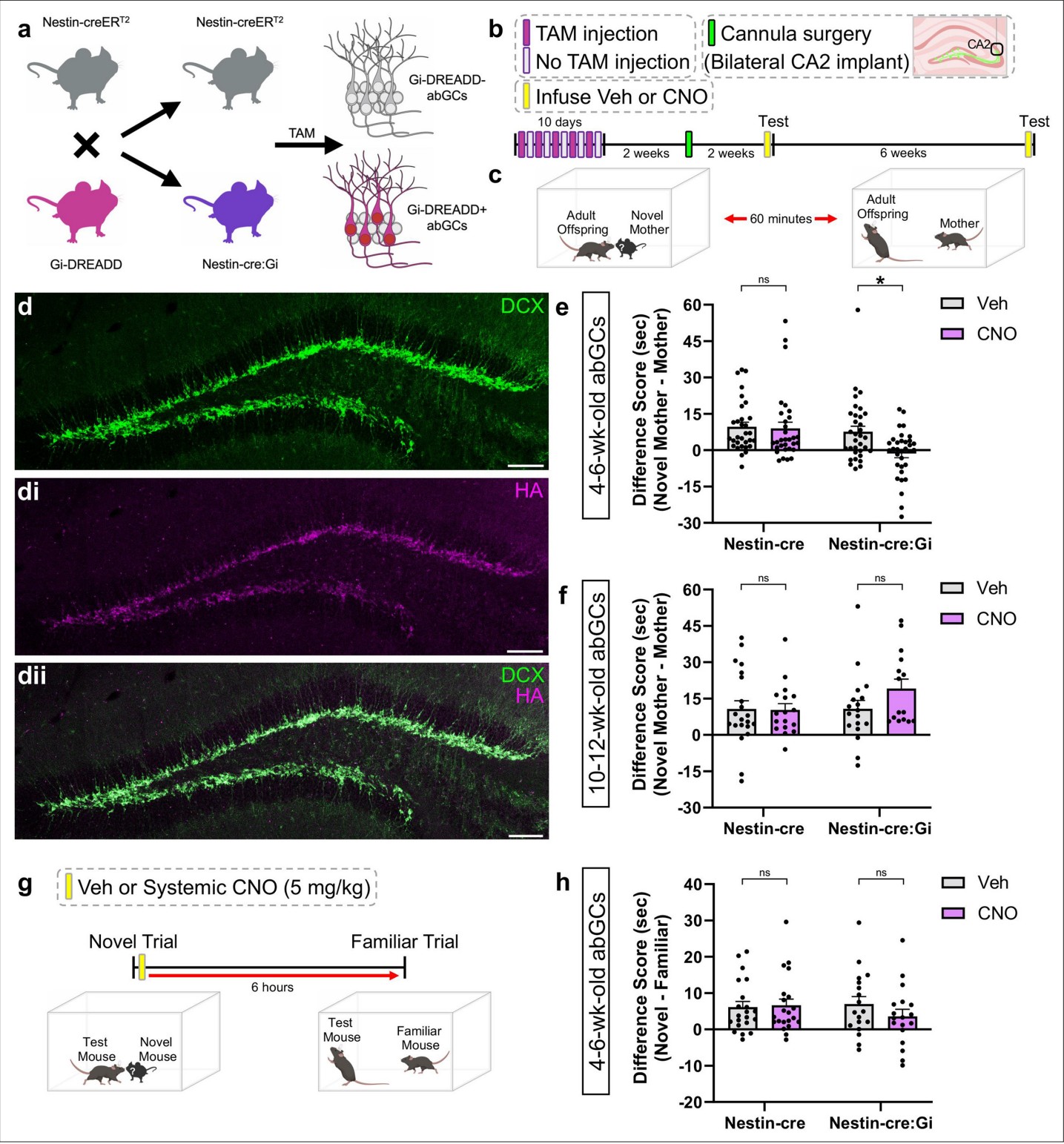

**Figure 2.** Inhibiting 4–6-week-old abGC projections to CA2 prevents discrimination between mothers and novel mothers. (**a**) Schematic of breeding to produce Nestin-cre:Gi double transgenic offspring. (**b**) Timeline outlining tamoxifen administration, cannula implantation, and behavioral testing. (**c**) Schematic demonstrating direct social interaction testing. (**d-dii**) Confocal images depicting 4–6-week-old Gi-DREADD + abGCs in the DG. (**e**) Inhibiting 4–6-week-old adult-born granule cell (abGC) projections to CA2 with clozapine-N-oxide (CNO) prevents discrimination between mother and novel mother (Nestin-cre: n=31; Nestin-cre:Gi: $n$ = 33; two-way RM ANOVA: Genotype x Drug: $F_{1,62}$ = 4.042, p=0.0487; Šídák's test p=0.0057). (**f**) Inhibiting 10–12-week old abGC projections to CA2 does not influence discrimination between mother and novel mother Nestin-cre: n=21 (Veh), n=17

*Figure 2 continued on next page*

*Figure 2 continued*

(CNO); Nestin-cre:Gi: n=18 (Veh), n=15 (CNO); Mixed-effects ANOVA: Genotype x Drug:$_{1,67}$ = 1.669, Pp=0.2008. (**g**) Schematic for experiment assessing abGC contributions to social memory consolidation. (**h**) Systemic inhibition of abGCs during memory consolidation has no significant effect on social memory (Nestin-cre: n=21; Nestin-cre:Gi: n=18; two-way RM ANOVA: Genotype x Drug: $F_{1,37}$ = 1.238, p=0.2730). *p<0.05, bars represent mean + SEM. ns = not significant. Scale bars in d=200 μm.

The online version of this article includes the following source data for figure 2:

**Source data 1.** Inhibiting 4–6-week old abGC projections to CA2 prevents discrimination between mothers and novel mothers.

supplement 2a). However, inhibition of abGCs in the presence of a social stimulus altered several features of SWR production, including decreasing SWR frequency, integral, and duration (*Figure 3— figure supplement 1a and b*; *Figure 3—figure supplement 2a*), but no effects on SWR peak amplitude (*Figure 3h*). Inhibition of abGCs during baseline recordings showed no change in SWR frequency, but an increase in SWR peak amplitude (*Figure 3i and j*). Conversely, inhibition of 10–12 week-old abGCs had no statistically significant effect on social novelty-induced increases in CA2 SWRs, nor did it affect other features of SWR production (*Figure 3k*, *Figure 3—figure supplements 1d, e and 2b*). These findings suggest that 4–6-week-old abGCs, but not 10–12-week-old abGCs, are necessary for promoting appropriate SWR responses during social novelty exploration and developmental social memory retrieval. In an additional study, we investigated if CA2 SWRs were modulated by the presence of a nonsocial stimulus (an object) (*Figure 3—figure supplement 3a–f*). Surprisingly, CA2 SWR generation was suppressed upon exposure to a nonsocial stimulus (*Figure 3—figure supplement 3b*). Inhibiting immature abGCs had no statistically significant effects on SWR features in response to a nonsocial stimulus (*Figure 3—figure supplement 3b–e*).

We next investigated abGC contributions to theta-gamma PAC (*Figure 4a*), which is increased after exposure to social stimuli (*Zhu et al., 2023*) and has been linked to nonsocial memory retrieval (*Tronel et al., 2015*). We found that exposure to the mother was associated with increased theta-mid-gamma PAC, and that this increase was prevented after inhibition of 4–6-week-old abGCs (*Figure 4b–d*). Conversely, 4–6-week-old abGCs had no influence on CA2 oscillation power across theta, low-gamma, and mid-gamma frequency bands (*Figure 4—figure supplement 1a–f*). Inhibition of a 10–12-week-old cohort of abGCs had no statistically significant effect on PAC signatures evoked during exposure to the mother (*Figure 4e–g*), although there appears to be an overall shift in gamma phase within theta cycles, especially during the mother exposure condition (*Figure 4g*). This suggests that abGC influence on CA2 network activity is limited to a specific abGC age range. Taken together with our SWR results, these findings suggest that 4–6-week-old abGCs support CA2 network oscillations present during both encoding and retrieval of social memories.

## Discussion

Hippocampal CA2 plays an important role in supporting social recognition (*Hitti and Siegelbaum, 2014*; *Smith et al., 2016*; *Laham et al., 2021*; *Diethorn and Gould, 2023*), yet the contributions of the abGC projection to this region have remained uninvestigated. Here, we show that ablation of abGCs impairs the retrieval of developmental memories of the mother. Recovery of abGCs and their projections to CA2 restored the ability to discriminate between novel mothers and caregiving mothers, revealing that abGCs support developmental social memory retrieval and not remote memory storage. Implementing a double transgenic mouse model, we found that chemogenetic inhibition targeted at the 4–6-week-old abGC projections to the CA2 prevents the ability to discriminate between mothers and novel mothers. This effect was not detected when inhibiting 10–12-week-old abGC projections, revealing that abGCs possess a time-sensitive ability to support retrieval of developmental mother memories.

It is important to note that we cannot rule out the possibility that projections from abGCs to the CA3 were also affected in double transgenic mice with CNO infusion, due to the potential diffusion of the drug away from the CA2 cannula site. Along these lines, studies have shown that the ventral CA3 is also involved in social recognition (*Chiang et al., 2018*) and that areas CA2 and CA3 may act together to coordinate replay (*Stöber et al., 2020*). Thus, the abGC-CA2 circuit may work in concert

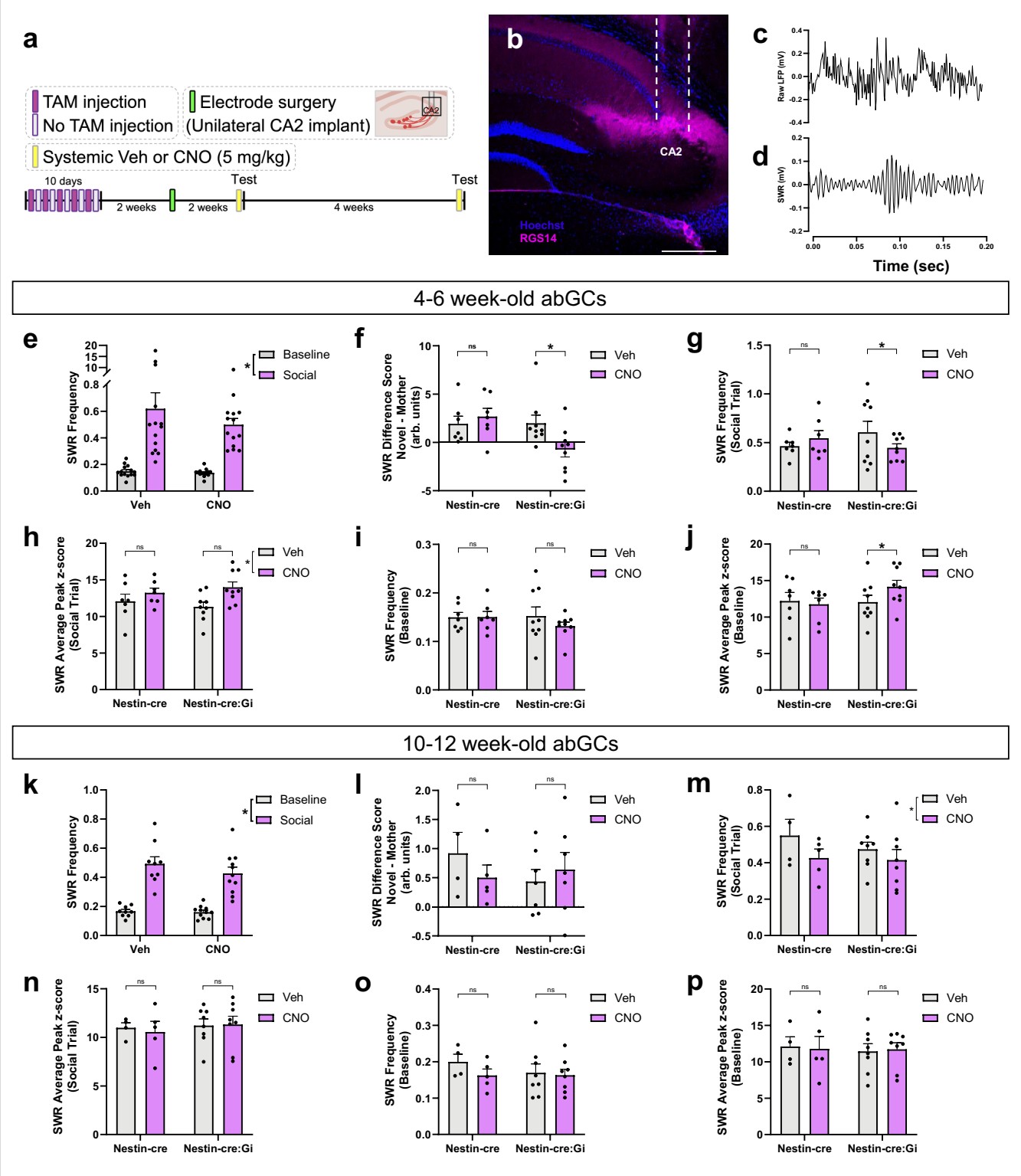

**Figure 3.** Adult-born neurons influence CA2 sharp wave ripple (SWR) changes associated with discrimination between novel mother and mother. (**a**) Timeline of experiment demonstrating tamoxifen injection schedule and electrode implantation in CA2. (**b**) Confocal image demonstrating accuracy of electrode placement. (**c**) Example of SWR trace and (**d**) filtered trace recorded from CA2. (**e**) CA2 SWR frequency is increased during social interactions, regardless of whether the stimulus is the novel mother or mother (Veh: n=14; CNO: n=14; 2-way RM ANOVA: Stimulus: $F_{1,13}$ = 46.40, p<0.0001). (**f**) Inhibiting 4–6-weekold adult-born granule cells (abGCs) with clozapine-N-oxide (CNO) prevents characteristic SWR production patterns present during exposure to novel mothers vs mothers (Nestin-cre: n=7; Nestin-cre:Gi: n=9; Mixed-effects ANOVA: Genotype x Drug: $F_{1,110}$ = 18.6529, p<0.0001;

*Figure 3 continued on next page*

*Figure 3 continued*

Tukey's test p<0.0001). (**g**) Inhibiting 4–6-week old abGCs significantly reduces social trial SWR production (Nestin-cre: n=7; Nestin-cre:Gi; n=9; Mixed-effects ANOVA: Genotype x Drug: $F_{1,228}$ = 16.2465, *P*<0.0001; Tukey's test p=0.0018). (**h**) Inhibiting 4–6 week old abGCs increases SWR peak amplitude during social interaction trials (Nestin-cre: n=7; Nestin-cre:Gi: n=9; Mixed-effects ANOVA: Genotype x Drug: $F_{1,238}$ = 3.69, p=0.0559) and (**j**) baseline (Nestin-cre: n=7; Nestin-cre:Gi: n=9; Mixed-effects ANOVA: Genotype x Drug: $F_{1,237}$ = 6.0322, p=0.01477; Tukey's test p=0.0133). (**i**) Changes in baseline SWR production did not reach significance (Nestin-cre: n=7; Nestin-cre:Gi: n=9; Mixed-effects ANOVA: Genotype x Drug: $F_{1,238}$ = 3.5101, p=0.0622). (**k**) At the 10–12 week abGC time point, CA2 SWR frequency is increased during social interactions, regardless of stimulus (Veh: n=9; CNO: n=11; Mixed-effects ANOVA: Stimulus: $F_{1,10}$ = 133.1, p<0.0001). (**l**) Inhibiting 10–12-week old abGCs has no influence on characteristic SWR production patterns present during exposure to the novel mother vs mother (Nestin-cre: n=4 (Veh), n=5 (CNO); Nestin-cre:Gi: n=7 (Veh and CNO); Mixed-effects ANOVA: Genotype x Drug: $F_{1,82}$ = 1.7431, p=0.1904), nor on (**m**) SWR production during social trials (Nestin-cre: n=4 (Veh), n=5 (CNO) Nestin-cre:Gi: n=8 (Veh and CNO); Mixed-effects ANOVA: Genotype x Drug: $F_{1,170}$ = 0.4496, p=0.503), or (**o**) during baseline recording (Nestin-cre: n=4 (Veh), n=5 (CNO); Nestin-cre:Gi: n=8 (Veh and CNO); Mixed-effects ANOVA: Genotype x Drug: $F_{1,170}$ = 2.0552, p=0.15352). (**n**) Inhibiting 10–12-week old adult-born granule cells (abGCs) has no influence on SWR peak amplitude social interaction trials (Nestin-cre: n=4 (Veh), n=5 (CNO); Nestin-cre:Gi: n=8 (Veh and CNO); Mixed-effects ANOVA: Genotype x Drug: $F_{1,18}$ = 0.9744, p=0.3250) or (**p**) during baseline (Nestin-cre: n=4 (Veh), n=5 (CNO); Nestin-cre:Gi: n=8 (Veh and CNO); Mixed-effects ANOVA: Genotype x Drug: $F_{1,172}$ = 0.540, p=0.8165), *p<0.05, bars represent mean + SEM. Scale bar in b=1500 μm. ns = not significant.

The online version of this article includes the following source data and figure supplement(s) for figure 3:

**Source data 1.** Adult-born neurons influence CA2 SWR changes associated with discrimination between novel mother and mother.

**Figure supplement 1.** 4–6-week-old a adult-born granule cells (bGCs) influence multiple features of CA2 sharp wave ripples (SWRs).

**Figure supplement 1—source data 1.** 4–6-weekold abGCs influence multiple features of CA2 SWRs.

**Figure supplement 2.** Normalized sharp wave ripple (SWR) generation.

**Figure supplement 2—source data 1.** Normalized sharp wave ripple (SWR) generation.

**Figure supplement 3.** Nonsocial stimuli suppress CA2 sharp wave ripple (SWR) frequency.

**Figure supplement 3—source data 1.** Nonsocial stimuli suppress CA2 sharp wave ripple (SWR) frequency.

with the abGC-CA3 circuit, followed by crosstalk between the two target areas to mediate social recognition.

We also investigated if abGCs support developmental mother memory retrieval via modulation of CA2 network activity and found that exposure to novel mothers produced a larger increase in CA2 SWR frequency than exposure to caregiving mothers. We also found the converse with CA2 theta/mid-gamma PAC, which was higher during exposure to caregiving mothers than to novel mothers. Both effects were significantly diminished after inhibiting 4–6-week-old abGCs. Thus, inhibition of immature abGCs alters SWR generation and theta-mid-gamma PAC in the CA2 region in a way that increases network activity similarity across novel and familiar social interactions. These changes likely affect broader hippocampal circuitry as inhibiting CA2 activity alters both SWRs and gamma oscillations in CA1 (*Alexander et al., 2018*; *Brown et al., 2020*). abGC inhibition of network orthogonalization may contribute to the inability to socially discriminate.

Our data suggest that abGCs contribute to the diversification of neuronal oscillatory states in the CA2 in the service of both recognizing social novelty and retrieving developmental social memories. The mechanisms underlying these effects remain unknown. Although we did not record single units along with neuronal oscillations, previous studies suggest that CA2 SWRs correspond with the activation of neuronal ensembles that are associated with previous social experience, at least during sleep (*Oliva et al., 2020*). Thus, by promoting the generation of SWRs during social, especially novel, experience, abGCs may provide a network conducive to replay. The ability of abGCs to facilitate the generation of SWRs and PAC may be related to their preferential connections to PV + interneurons, which have been causally linked to SWRs, as well as to theta and gamma oscillations (*Schlingloff et al., 2014*; *Stark et al., 2014*; *Amilhon et al., 2015*; *Gan et al., 2017*; *Antonoudiou et al., 2020*; *Park et al., 2020*; *Kriener et al., 2022*).

A time-limited role for abGCs in nonsocial memory has been previously reported (*Gu et al., 2012*; *Song et al., 2012*; *Swan et al., 2014*; *Kropff et al., 2015*). Our results raise questions about how these cells change as they mature, as well as what the function of abGC projections to the CA2 might be after their involvement in remote mother memory retrieval has ended. Previous studies have shown that abGCs are highly excitable and exhibit enhanced synaptic plasticity 4–6 weeks after their

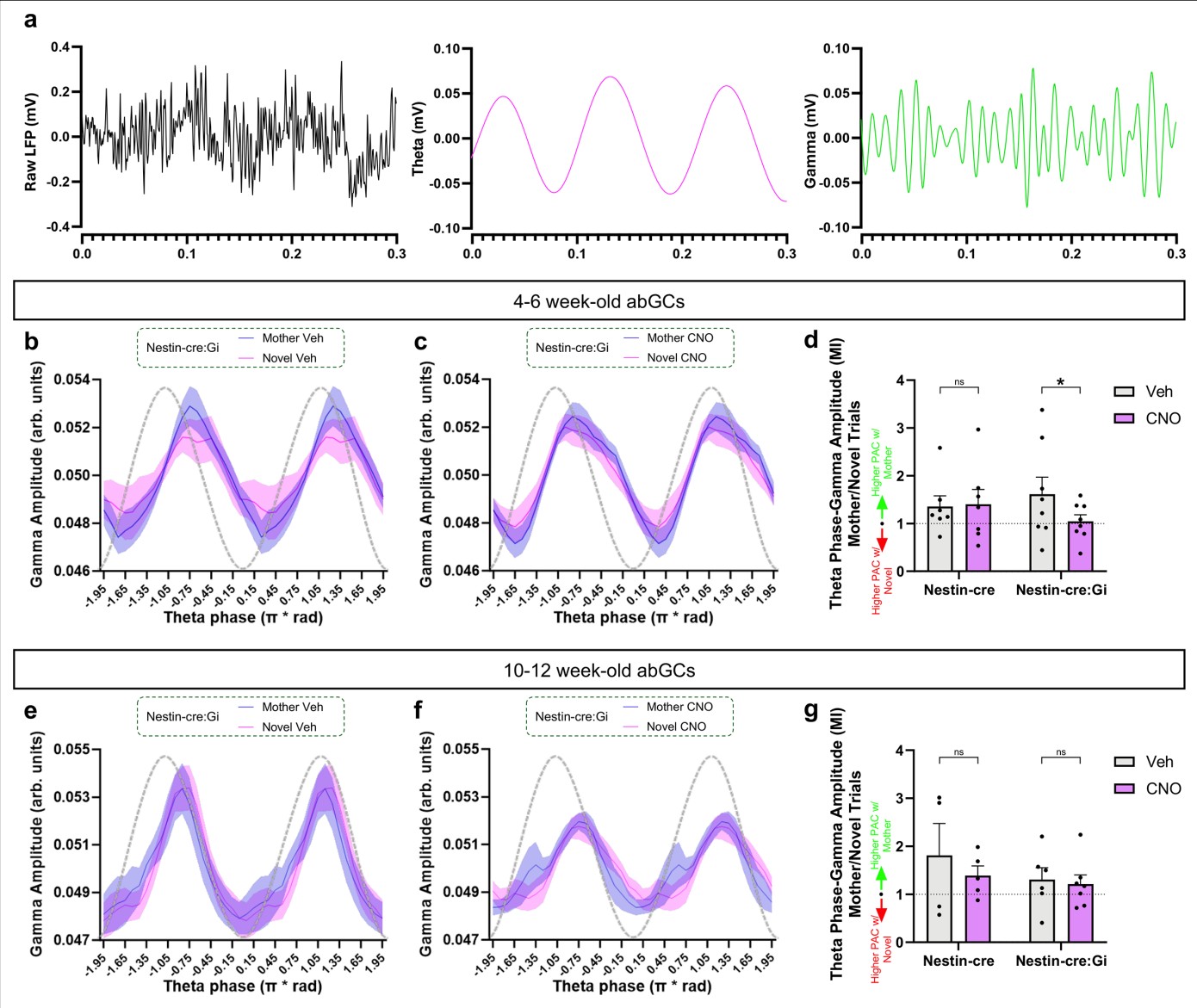

**Figure 4.** Adult-born neurons influence CA2 phase-amplitude coupling (PAC) changes associated with retrieval of developmental memories of the mother. (**a**) Example trace of raw local field potential (LFP) (black), 4–12 Hz theta oscillation (magenta), and 50–100 Hz gamma oscillation (green). (**b,c**) Graphs demonstrating mid-gamma amplitude modulation across theta phases in Nestin-cre:Gi mice. (**d**) Inhibiting 4–6-week old adult-born granule cells (abGCs) abolishes the increase in theta-mid-gamma PAC present during exposure to the mother (Nestin-cre: n=7; Nestin-cre:Gi: n=8; Mixed-effects ANOVA: Genotype x Drug: $F_{1,103}$ = 4.6729, p=0.03296; Šídák's test p=0.0219). (**e,f**) Graphs demonstrating mid-gamma amplitude modulation across theta phases in Nestin-cre:Gi mice. (**g**) Inhibiting 10–12-week old abGCs has no influence on theta-mid-gamma PAC during exposure to the mother (Nestin-cre: n=4 (Veh), n=5 (CNO); Nestin-cre:Gi: n=8 (Veh), n=7 (CNO); Mixed-effects ANOVA: Genotype x Drug: $F_{1,18}$ = 0.2616, p=0.6153). *p<0.05, bars represent mean + SEM.

The online version of this article includes the following source data and figure supplement(s) for figure 4:

**Source data 1.** Adult-born neurons influence CA2 phase-amplitude coupling (PAC) changes associated with retrieval of developmental memories of the mother.

**Figure supplement 1.** 4–6-week old adult-born granule cell (abGC) inhibition does not influence theta and gamma power during social interaction.

**Figure supplement 1—source data 1.** 4–6-week old adult-born granule cell (abGC) inhibition does not influence theta and gamma power during social interaction.

generation, and that these properties appear to be diminished by 8 weeks post-mitosis (*Song et al., 2012*; *Toni and Schinder, 2015*). As abGCs mature and become more inhibited, they may function to provide input to the CA2 only when social stimuli are especially salient, such as when they are threatening or associated with a potent reward. It may also be relevant to determine whether abGC inputs to the CA2 affect synaptic input from the supramammillary nucleus of the hypothalamus, which is known to respond to social novelty (*Chen et al., 2020*) and drive novelty-induced increases in adult neurogenesis (*Li et al., 2022*). It is possible that abGCs influence this circuit by increasing excitatory tone in the same target cells, which in turn alters neuronal oscillations, although this hypothesis remains untested.

# Materials and methods

**Key resources table**

| Reagent type (species) or resource | Designation | Source or reference | Identifiers | Additional information |
|---|---|---|---|---|
| Strain, strain background (*Mus musculus* - males and females) | C57Bl/6-Tg(Nes-cre/ERT2)KEisc/J, | Jackson labs | 016261 | |
| Strain, strain background (*Mus musculus* - males and females) | B6.129-*Gt(ROSA)26Sortm1(CAG-CHRM4\*,-mCitrine)Ute*/J, | Jackson labs | 026219 | |
| Strain, strain background (*Mus musculus* - males and females) | C57Bl/6, | Jackson labs | 000644 | |
| Strain, strain background (*Mus musculus* - males and females) | B6.129P2-*Pvalbtm1(cre)Arbr*/J, | Jackson labs | 017320 | |
| Antibody | 3R-Tau, mouse, monoclonal | Millipore Sigma | 05–803 | 1:500 |
| Antibody | Znt3, rabbit, polyclonal | Alomone Labs | AZT-013 | 1:500 |
| Antibody | RGS14, mouse, monoclonal | UC Davis/NIH Neuromab | 75–170 | 1:500 |
| Antibody | PCP4, rabbit, polyclonal | Sigma-Aldrich | HPA005792 | 1:500 |
| Software, algorithm | FIJI | ImageJ | NIH | |
| Software, algorithm | Prism 9.2.0 | Graphpad | GraphPad Software | |
| Software, algorithm | NeuroExplorer | NEX5 | NEX Technologies | |
| Other | Cre-dependent mCherry AAV2/5, pAAV-hSyn-DIO-mCherry | Addgene | 50459 | Virus – see surgical procedures in Methods |

## Animals

All animal procedures were approved by the Princeton University Institutional Animal Care and Use Committee and were in accordance with the National Research Council Guide for the Care and Use of Laboratory Animals. C57BL/6 male and female mice were obtained from Jackson laboratories and were used for immunolabeling. Transgenic mice expressing herpes simplex virus thymidine kinase (TK) under the GFAP promoter were bred in the Princeton Neuroscience Institute animal colony with founders provided by Dr. Heather Cameron at the National Institute of Mental Health for adult neurogenesis ablation studies. GFAP-TK mice were generated by crossbreeding CD1 male mice with heterozygous GFAP-TK female mice (*Snyder et al., 2011*). Nestin-CreERT2 and R26LSL-hM4Di single transgenic mice were obtained from Jackson labs and bred in the Princeton Neuroscience Institute as single or double transgenic mice for cannula implantation and electrophysiology experiments. PV-cre mice were obtained from Jackson labs and bred in the Princeton Neuroscience Institute to investigate abGC projections to CA2 PV + interneurons. All studies used mixed-sex groups. Transgenic mice were genotyped by Transnetyx from ear punches at P15 using real-time PCR, separated from their dams at P21, and housed in same-sex groups. GFAP-TK and CD1 mice were treated with VGCV starting around P60. Nestin-cre and nestin-cre:Gi transgenic mice were subjected to craniotomy for bilateral

cannula implantation, or unilateral electrode implantation starting around P60. GFAP-TK mice were group housed by genotype as well as sex to avoid any potential behavioral differences after ablation of abGCs (*Tsuda et al., 2023*), while single and double transgenic animals were group-housed with mixed genotypes. Animals were housed in Optimice cages on a reverse 12/12 hr light/dark cycle.

## Behavioral testing

The direct social interaction test was used (*Laham et al., 2021*) to determine whether adult offspring were able to discriminate between their mother and a novel mother. Novel mothers were female mice of the same age and reproductive experience as the mothers, which the adult offspring had never encountered. To avoid the confound of altered social preference during times of sexual receptivity, the estrous cycle was tracked in the mother and novel mother, and behavioral testing only occurred when stimulus mice were in diestrus (*Cora et al., 2015*). Adult GFAP-TK, Nestin-cre, and Nestin-cre:Gi offspring underwent a social interaction test in which they directly interacted with the mother, or a novel mother for 5 min. After a 1 hr delay spent in the home cage with cage mates, mice were introduced to the stimulus mouse not previously encountered for 5 min. The order of stimulus mouse exposure was counterbalanced in all tests. All behavioral tests were recorded, and a trained observer scored behavior testing under blind conditions. Testing apparatuses were cleaned with 70% ethanol after each trial. Bouts of investigation were characterized as direct sniffing of the stimulus mouse's anogenital region, body, and head.

An additional experiment explored abGC contributions to social memory consolidation. Nestin-cre and Nestin-cre:Gi mice were placed in a testing apparatus with a novel adult mouse and were permitted to investigate for 5 min. Immediately after the interaction, mice received a systemic injection of Veh or 5 mg/kg CNO and were returned to their cages in the vivarium for 6 hr. At the conclusion of the 6 hr intertrial interval, mice were returned to the testing apparatus housing the stimulus mouse they had previously encountered and allowed to investigate for an additional 5 min. Previous studies have shown that 6 hr after CNO treatment, other types of behavioral changes return to baseline, and baseline neuronal activity is largely restored (*Alexander et al., 2009*; *Whissell et al., 2016*; *Ray et al., 2011*). For each experiment that involved a social interaction measure, the time spent moving was recorded. Total locomotion was divided by trial duration to create a % locomotion measure.

## Surgical procedures

Mice were deeply anesthetized with isoflurane (2–3%) and placed in a stereotaxic apparatus (Kopf) under a temperature-controlled thermal blanket for all surgeries. The head was leveled using bregma, lambda, and medial-lateral reference points before virus injection or implantation of cannula or electrode. PV-cre mice received bilateral injections of a cre-dependent mCherry AAV into CA2 (AP: –1.82, ML:+/-2.15, DV: –1.67) through a WPI nanofil 33-gauge beveled needle. Nestin-cre and Nestin-cre:Gi mice were bilaterally implanted with cannula (Plastics One, Cat# C315GS-5/SP) targeting CA2. Dummy cannula (Plastics One, Cat# C315DCS5/SPC) were tightened inside guides prior to implantation. After lowering to the desired target region, cannula were secured in place using metabond followed by dental cement (Bosworth Trim). For CA2 recordings, Nestin-cre and Nestin-cre:Gi mice were implanted unilaterally with a custom-made 4-wire electrode array (Microprobes) into the right hemisphere targeting CA2. 4 bone screws were implanted into the skull with one screw positioned on the contralateral hemisphere serving as ground. The ground wire was wrapped around the ground screw and covered with metallic paint to ensure maximum contact. Electrode implants were kept in place using metabond followed by dental cement (Bosworth Trim). Two weeks after surgery, cannula mice were infused with either vehicle or CNO (see Drug treatments) before being tested on behavioral tasks (see Behavioral testing); electrode mice received systemic injections of vehicle or CNO 30 min prior to electrophysiological recordings (see Electrophysiology recordings).

## Drug treatments

Administration of the antiviral drug valganciclovir (VGCV) ablates adult neurogenesis in GFAP-TK animals. CD1 and GFAP-TK mice underwent behavioral testing at three time points. Animals first underwent testing prior to VGCV administration (VGCV- trial). After the first round of behavioral testing, VGCV was added to powdered rodent chow (227 mg of VGCV per kg chow) for 6 weeks. After 6 weeks of consuming VGCV chow, animals underwent behavioral testing for a second time (VGCV +

trial). At the conclusion of the VGCV + trial, VGCV chow was removed from all cages and replaced with standard chow. After 6 weeks of standard chow consumption, animals underwent a third round of behavioral testing (VGCV- recovery). Mice were perfused shortly after the third round of behavioral testing to assess the extent of adult neurogenesis recovery.

Cannula-implanted Nestin-cre and Nestin-cre:Gi mice underwent social discrimination testing (with novel mother and mother stimulus mice) twice, once after vehicle cannula infusion and once after CNO cannula infusion. The order of drug administration (vehicle or CNO) was counterbalanced across groups. Electrode-implanted mice underwent direct social interaction testing with systemic vehicle or CNO administration. Mice were given a 48 hr minimum rest period between vehicle and CNO tests. 30 min prior to the first stimulus mouse exposure, test mice received vehicle or CNO cannula infusions. For cannula infusions, 200 nl of vehicle or CNO (2 µg/µl of CNO dissolved in DMSO suspended in saline) (*Chang and Gean, 2019*) was infused per hemisphere over 1 min into CA2 using a syringe pump (Harvard Apparatus) mounted with a 1 µl syringe (Hamilton). The internal cannula remained in place for 1 additional minute after the infusion was completed to allow for diffusion of the drug.

## Electrophysiology recordings

Local field potentials (LFPs) were recorded using a wireless head stage (TBSI, Harvard Biosciences). To habituate mice to the weight of the recording head stage, mice were connected to a custom head stage with equivalent weight while in the home cage for 10 min a day for five consecutive days. The behavioral testing paradigm consisted of a 3 min baseline followed by a 5 min social interaction period. Animals underwent this recording paradigm for both mother and novel mother trials. Vehicle or 5 mg/kg CNO was administered via IP injection 30 min before testing. In an additional experiment, Nestin-cre and Nestin-cre:Gi mice with CA2 electrodes were recorded during a 1 min baseline followed by a 2 min nonsocial object exposure trial. The nonsocial stimulus was a plastic toy animal of similar size to the mouse. Both baseline and nonsocial stimulus trials took place in an apparatus identical to that of the social stimulus experiments. The nonsocial stimulus trials were conducted after the conclusion of all social testing at the 4–6-week-old abGC time point. LFPs were recorded continuously throughout baseline and stimulus trials. The data were transmitted to a wireless receiver (Triangle Biosystems) and recorded using NeuroWare software (Triangle Biosystems).

## Sharp wave ripple analysis

All recordings were processed using Neuroexplorer software (Nex Technologies) and custom Python scripts (*Laham and Zahn, 2023*). For SWR analyses, continuous LFP data were notched at 60 Hz and band-pass filtered between 140 and 220 Hz. Signal underwent Hilbert transform before being z-scored. Using a custom Python script, SWRs were considered as events exceeding three standard deviations for a minimum of 15 ms. SWRs occurring within 15 ms of one another were merged into a single SWR event. SWR frequency, peak amplitude, integral, duration, and raw number were normalized to the respective baseline trial. SWR integral, which provides information about the average total power of SWR events and has been shown to influence SWR propagation (*De Filippo and Schmitz, 2023*) was calculated by summing z-scored amplitude values from beginning to the end of each SWR envelope, and then averaging these values across all summed envelopes. SWR measures were determined for baseline, novel mother, and mother trials for the entirety of each testing period.

## Phase-amplitude coupling analysis

Theta (4–12 Hz), low-gamma (25–50 Hz), and mid-gamma (50–100 Hz) oscillations were extracted from the raw LFP signal obtained during the entire baseline, novel mother exposure, and mother exposure recordings using a custom Python script. Theta and mid-gamma recordings underwent a Hilbert transform, and theta phase and mid-gamma analytic signal were stored. PAC was quantified using a Modulation Index (*Tort et al., 2010*).

## Histology

Mice were deeply anesthetized with Euthasol (Virbac) and were transcardially perfused with cold 4% paraformaldehyde (PFA). Extracted brains were postfixed for 48 hr in 4% PFA at 4°C followed by an additional 48 hr in 30% sucrose at 4°C for cryoprotection before being frozen in cryostat embedding medium at –80°C. Hippocampal coronal sections (40 µm) were collected using a cryostat (Leica).

Sections were blocked for 1.5 hr at room temperature in a PBS solution that contained 0.3% Triton X-100 and 3% normal donkey serum. Sections were then incubated overnight while shaking at 4°C in the blocking solution that contained combinations of the following primary antibodies: mouse anti-3-repeat-tau protein (3R-Tau, 1:500, Millipore, Cat# 05–803), rabbit anti-Purkinje cell protein 4 (PCP4, 1:500, Sigma-Aldrich, Cat# HPA005792), rabbit anti-zinc transporter 3 (ZnT3, 1:500, Alomone labs, Cat# AZT-013). For 3R-Tau immunohistochemistry, sections were subjected to an antigen retrieval protocol that involved incubation in sodium citrate and citric acid buffer for 30 min at 80°C prior to blocking solution incubation. Washed sections were then incubated for 1.5 hr at room temperature in secondary antibody solutions that contained combinations of the following secondaries: donkey anti-rat Alexa Fluor 568 (1:500, Abcam), donkey anti-mouse Alexa Fluor 568 or 647 (1:500, Invitrogen), or donkey anti-rabbit Alexa Fluor 488 (1:500, Invitrogen).

Washed sections were then counterstained with Hoechst 33342 for 10 min (1:5000 in PBS, Molecular Probes), mounted onto slides, and coverslipped with Vectashield (Vector labs). Slides were coded until the completion of the data analysis. Sections through the dorsal hippocampus from cannula and electrophysiology studies were stained for Hoechst 33342 to verify accurate cannula and electrode placement.

## Optical intensity measurements

Z-stack images of the CA2 and corpus callosum were taken using a 40 x objective with a 0.5 µm step size through the entire 40 µm stack on a Leica SP8 confocal with LAS X software (version 35.6). The CA2 was defined by PCP4 labeling. Collected z-stack images were analyzed for optical intensity in Fiji (NIH).

For high magnification (40 X objective with 3 X zoom) intensity analyses of 3R-Tau labeled puncta in close proximity to dendrites of mCherry + PV cells and dendrites of PCP4 + pyramidal cells, mCherry + and PCP4 + labeled cell bodies were first identified in the pyramidal cell layer. Then, proximal dendrites were identified emanating from selected labeled cells and the ROI was traced around the dendrite. The dendrite was defined from the initial tapering off from the cell body and extending for 7 µm in length, which was the maximum that could be traced to maintain certainty that the entirety of the selected dendrite belonged to a specific cell body. All dendrites selected for analysis were located either in the pyramidal cell layer or the stratum lucidum of CA2. For each brain, 10 labeled cells of each cell type (PV + cells or PCP4 + pyramidal cells) were selected, and from these, one dendrite per cell was randomly selected for analysis per brain. The intensity of 3R-Tau labeling on dendrites was determined for each cell type using Fiji as follows: A background subtraction using a rolling ball radius (50 pixels) was applied to the image stacks. A region of interest (ROI) was drawn, and the mean gray value was collected throughout the image stack. Averages were obtained for each brain and statistical comparisons were made between intensity on PV-mCherry + proximal dendrites and intensity on PCP4 + proximal dendrites.

For overall intensity values of 3R-Tau to assess abGC innervation of CA2, the ROI included the pyramidal cell layer and stratum lucidum. The mean gray value of the ROI was calculated for each z-slice and the maximum mean gray value for each z-stack was taken from three sections per brain. The maximum of the CA2 ROI was divided by the maximum of the corpus callosum ROI for each section. Averages were obtained for each brain and statistical comparisons were made between lower magnification 3R-Tau intensity across different GFAP-TK groups.

## Cell density and percentage measurements

3R-Tau + cell bodies were counted in the dorsal dentate gyrus of the hippocampus on four neuroanatomically matched sections using an Olympus BX-60 microscope with a 100 x oil objective. The area measurements were collected using Stereo Investigator software (MBF). The density of 3R-Tau+ cells was determined by dividing the total number of positively labeled cells by the volume of the subregion (ROI area multiplied by 40 µm section thickness).

## Statistical analyses

Statistical analyses are presented in the figure legends. For histological analyses, datasets were analyzed using an unpaired two-tailed Student's t-test or Mann-Whitney U tests. For behavioral analyses involving two group comparisons, data sets were analyzed using either unpaired two-tailed

Student's t-tests or a repeated measures two-way ANOVA, as appropriate. For behavioral analyses involving virus manipulations, data sets were analyzed using either a two-way ANOVA, a mixed-effects model, or a repeated measures three-way ANOVA with Šídák or Tukey post hoc tests. Because electrophysiological measurements were taken from multiple electrodes within each mouse, these data were analyzed with linear mixed-effects ANOVAs using the lme4 package (*Bates et al., 2015*). The level of the measurement was explained by drug, virus, genotype, the two-way interaction, the three-way interaction, and a random effect of mouse. Tukey post hoc comparisons were used to follow up any significant main effects or interactions using the emmeans package (*Lenth, 2022*). All data sets are expressed as the mean ± SEM on the graphs and statistical significance was set at $p < 0.05$ with 95% confidence. GraphPad Prism 9.2.0 (GraphPad Software), Excel (Microsoft), or R studio were used for statistical analyses. All graphs were prepared using GraphPad Prism 9.2.0 (GraphPad Software).

## Acknowledgements

This work was supported by the National Institutes of Health, NIMH 1R01MH118631-01 to EG. The authors thank Samantha H Wang for performing surgeries on PV-cre mice and for help with editing the text. The authors acknowledge Biorender for assistance with the figure schematics.

## Additional information

### Funding

| Funder | Grant reference number | Author |
|---|---|---|
| National Institute of Mental Health | MH118631-01 | Elizabeth Gould |

The funders had no role in study design, data collection and interpretation, or the decision to submit the work for publication.

### Author contributions

Blake J Laham, Conceptualization, Data curation, Formal analysis, Investigation, Visualization, Writing – original draft, Writing – review and editing; Isha R Gore, Formal analysis, Investigation, Visualization, Writing – original draft, Writing – review and editing; Casey J Brown, Data curation, Formal analysis, Investigation, Visualization, Writing – original draft, Writing – review and editing; Elizabeth Gould, Conceptualization, Data curation, Supervision, Funding acquisition, Writing – review and editing

### Author ORCIDs

Elizabeth Gould (iD) http://orcid.org/0000-0002-8358-0236

### Ethics

This study was carried out in accordance with the recommendations in the Guide for the Care and Use of Laboratory Animals of the National Institutes of Health. Mice were treated according to approved Princeton University institutional animal care and use committee (IACUC) protocol (#1857).

Reviewer #2 (Public review): https://doi.org/10.7554/eLife.90600.3.sa1
Reviewer #3 (Public review): https://doi.org/10.7554/eLife.90600.3.sa2
Author response https://doi.org/10.7554/eLife.90600.3.sa3

## Additional files

### Supplementary files
• MDAR checklist

### Data availability

All data generated during this study are included in the manuscript and supporting files; source data files have been provided for all figures. The raw electrophysiology and histology data generated in this

study are available in the Dryad database: https://doi.org/10.5061/dryad.rbnzs7hhj. Custom pPython script is available for both SWR and PAC analyses: https://github.com/einaraz/SharpWaveRipple/ (copy archived at *Laham and Zahn, 2023*).

The following dataset was generated:

| Author(s) | Year | Dataset title | Dataset URL | Database and Identifier |
|---|---|---|---|---|
| Laham BJ | 2024 | Adult-born granule cells modulate CA2 network activity during retrieval of developmental memories of the mother | https://doi.org/10.5061/dryad.rbnzs7hhj | Dryad Digital Repository, 10.5061/dryad.rbnzs7hhj |

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
