## [Editor Report · eLife assessment]

This paper reports a **valuable** set of new results. The main result is that the projection from adult-born granule cells in the dentate gyrus to the hippocampal subfield CA2 is necessary for the retrieval of a social memory formed during development, and **solid** evidence is provided to support this conclusion.

---

## [Referee Report · Reviewer #2 (Public review)]

Summary:

Laham et al. investigate how the projection from adult born granule cells into CA2 affects the retrieval of social memories at various developmental points. They use chemogenetic manipulations and electrophysiological recordings to test how this projection affects hippocampal network properties during behavior. The study is of relevant interest for the neuroscience community and the results are important for our understanding of how social memories of different nature (remote or immediate) are encoded and supported by the hippocampal circuitry. The behavioral experiments after abGC projections to CA2 are compelling as they show clearly distinct behavioral readout. While the electrophysiological experiments are difficult to interpret without more single cell responses quantifications, they clearly show that more than one region in the hippocampus is involved in the formation of social memories.

---

## [Referee Report · Reviewer #3 (Public review)]

Laham et al. present a manuscript investigating the function of adult-born granule cells (abGCs) projecting to the CA2 region of the hippocampus during social memory. It should be noted that no function for the general DG to CA2 projection has been proposed yet. The authors use targeted ablation, chemogenetic silencing and in vivo ephys to demonstrate that the abGCs to CA2 projection is necessary for the retrieval of a remote social memories such as the memory of one's mother. They also use in vivo ephys to show that abGCs are necessary for differential CA2 network activity, including theta-gamma coupling and sharp wave-ripples, in response to novel versus familiar social stimuli.

The question investigated is important since the function of DG to CA2 projection remained elusive a decade after its discovery. Overall, the results are interesting but focused to the social memory of the mother and their description in the manuscript and figures is too cursory. For example, raw interaction times must be shown before their difference. The assumption that mice exhibit social preference between familiar or novel individuals such as mother and non-mother based on social memory formation, consolidation and retrieval should be better explained throughout the manuscript. Thus, when describing the results, the authors should comment on changes in preference and how this can be interpreted as a change in social memory retrieval. Several critical experimental details such as the total time of presentation to the mother and non-mother stimulus mice are also lacking from the manuscript. The in vivo e-phys results are interesting as well but even more succinct with no proposed mechanism as to how abGCs could regulate SWR and PAC in CA2.

The manuscript is well-written with the appropriate references. The choice of behavioral test is somewhat debatable however. It is surprising the authors chose to use a direct presentation test (presentation of the mother and non-mother in alternance) instead of the classical 3-chamber test which is particularly appropriate to investigate social preference. Since the authors focused exclusively on this preference, the 3-chamber test would have been more adequate in my opinion. It would greatly strengthened the results if the authors could repeat a key experiment from their investigation using such test. In addition, the authors only impaired the mother's memory. An additional experiment showing that disruption of the abGCs to CA2 circuit impairs social memory retrieval in general would allow to generalize the findings to social memories in general. As the manuscript stands, the authors can only conclude as to the importance of this circuit for the memory of the mother. Developmental memory implies the memory of familiar kin as well.

The in vivo ephys section (Figure 3) is interesting but even more minimalistic and it is unclear how abGCs projection to CA2 can contribute to SWR and theta-gamma PAC. In figure 1, the authors suggest that abGCs project preferentially to PV+ neurons in CA2. At minima, the authors should discuss how the abGCs to PV+ neurons to CA2 pyramidal neurons circuit can facilitate SWR and theta-gamma PAC.

Finally, proposing a function for 4-6-week-old abGCs projecting to CA2 begs two questions: What are abGCs doing once they mature further and more generally, what is the function of the DG to CA2 projection? It would be interesting for the authors to comment on these questions in the discussion.

Revision:

The authors have followed my recommendations except for the ones suggesting new experiments. As a result, the clarity of the manuscript and the links between evidence and claims have improved by the message is quite reduced. Many important questions remain open such as: What makes mother's memories so special they require the abGC projection to CA2 unlike other types of social memories? Do abGCs truly connect CA2 PV+ interneurons and how does this connection shape sharp-wave ripples in CA2?

---

## [Author Response]

The following is the authors’ response to the original reviews.

**Public Reviews:**

**Reviewer #1:**
Summary:This manuscript provides some valuable findings concerning the hippocampal circuitry and the potential role of adult-born granule cells in an interesting long-term social memory retrieval. The behavior experiments and strategy employed to understand how adult-born granule cells contribute to long-term social discrimination memory are interesting.

We thank the reviewer for the positive evaluation.

I have a few concerns, however with the strength of the evidence presented for some of the experiments. The data presented and the method described is incomplete in describing the connection between cell types in CA2 and the projections from abGCs. Likewise, I worry about the interpretation of the data in Figures 1 and 2 given the employed methodology. I think that the interpretation should be broadened. This second concern does not impact the interest and significance of the findings.

In response to this concern, we have removed the data concerning abGC projections to PCP4+ and PV-GFP+ cell bodies from Figure 1 and have focused this analysis on dendrites. We now provide high magnification images of dendrites and expand on the methodology, results, and interpretations in the manuscript. We also broaden the interpretation throughout the manuscript to address the reviewer’s concern.

Strengths:The behavior experiments are beautifully designed and executed. The experimental strategy is interesting.

We appreciate these positive comments.

Weaknesses:The interpretation of the results may not be justified given the methods and details provided.

We have addressed this concern by providing more methodological details and broadening our interpretation of the results.

Reviewer #2:Summary:Laham et al. investigate how the projection from adult-born granule cells into CA2 affects the retrieval of social memories at various developmental points. They use chemogenetic manipulations and electrophysiological recordings to test how this projection affects hippocampal network properties during behavior. I find the study to be very interesting, the results are important for our understanding of how social memories of different natures (remote or immediate) are encoded and supported by the hippocampal circuitry. I have some points that I added below that I think could help clarify the conclusions:

We appreciate the positive assessment and have addressed the more specific points below.

My major concern with the manuscript was that making the transitions between the different experiments for each result section is not very smooth. Maybe they can discuss a bit in a summary conclusion sentence at the end of each result section why the next set of experiments is the most logical step.

In response, we have added summary conclusion sentences at the end of each result section.

In line 113, the authors say that "the DG is known to influence hippocampal theta-gamma coupling and SWRs". Another recent study Fernandez-Ruiz et al. 2021, examined how various gamma frequencies in the dentate gyrus modulate hippocampal dynamics.

We cite this paper in the revised manuscript.

Having no single cells in the electrophysiological recordings makes it difficult to interpret the ephys part. Perhaps having a discussion on this would help interpret the results. If more SWRs are produced from the CA2 region (perhaps aided by projections from abGC), more CA2 cells that respond to social stimuli (Oliva et al. 2020) would reactivate the memories, therefore making them consolidate faster/stronger. On the other hand, the projections from abGC that the authors see, also target a great deal of PV+ interneurons, which have been shown to pace the SWRs frequency (Stark et al 2014, Gan et al 2017), which further suggests that this projection could be involved in SWRs modulation.

We discuss these possibilities and cite Gan et al 2017, Schlingloff et al., 2014, and Stark et al., 2014 in the revised manuscript.

The authors should cite and discuss Shuo et al., 2022 (A hypothalamic novelty signal modulates hippocampal memory).

We mention Chen et al (A hypothalamic novelty signal modulates hippocampal memory.) in the revised manuscript. “Shuo” is the first name of the first author on this paper, so we believe that this is the same paper to which the reviewer refers.

I think the authors forgot to refer to Fig 3a-f, maybe around lines 163-168.

We thank the reviewer for pointing out this error. In the revised manuscript, we refer to all figure panels. Since Fig 3 is now broken into two figures (Fig 3 and 4), the panel lettering has changed in the revised manuscript.

Are the SWRs counted only during interaction time or throughout the whole behavior session for each condition?

The SWRs are counted throughout the whole behavior session for each condition. This is now stated in the revised manuscript.

Figure 3t shows a shift in the preferred gamma phase within theta cycles as a result of abGC projections to CA2 ablation with CNO, especially during Mother CNO condition. I think this result is worth mentioning in the text.

We now mention this finding in the revised manuscript.

Figure 3u in the legend mention "scale bars = 200um", what does this refer to?

The scale bar refers to that shown in Figure 3b, which is now indicated in the legend.

What exactly is calculated as SWR average integral? Is it a cumulative rate? Please clarify.The integral measure provides information regarding the average total power of SWR events. It sums z-scored amplitude values from beginning to the end of each SWR envelope, and then takes the average across all summed envelopes. SWR integral has been shown to influence SWR propagation (De Filippo and Schmitz, 2023). This is now described in the text.Alexander et al 2017, "CA2 neuronal activity controls hippocampal oscillations and social behavior", examined some of the CA2 effects in the hippocampal network after CNO silencing, and the authors should cite it.

Alexander et al., 2018, which we believe is the relevant paper, is now cited in the revised manuscript.

Strengths:Behavioral experiments after abGC projections to CA2 are compelling as they show clearly distinct behavioral readout.

We thank the reviewer for this positive assessment.

Weaknesses:Electrophysiological experiments are difficult to interpret without additional quantifications (single-cell responses during interactions etc.)

We have addressed this concern by expanding the interpretation of our results.

**Reviewer #3:**
Laham et al. present a manuscript investigating the function of adult-born granule cells (abGCs) projecting to the CA2 region of the hippocampus during social memory. It should be noted that no function for the general DG to CA2 projection has been proposed yet. The authors use targeted ablation, chemogenetic silencing, and in vivo ephys to demonstrate that the abGCs to CA2 projection is necessary for the retrieval of remote social memories such as the memory of one's mother. They also use in vivo ephys to show that abGCs are necessary for differential CA2 network activity, including theta-gamma coupling and sharp wave-ripples, in response to novel versus familiar social stimuli.The question investigated is important since the function of DG to CA2 projection remained elusive a decade after its discovery. Overall, the results are interesting but focused on the social memory of the mother, and their description in the manuscript and figures is too cursory. For example, raw interaction times must be shown before their difference. The assumption that mice exhibit social preference between familiar or novel individuals such as mother and non-mother based on social memory formation, consolidation, and retrieval should be better explained throughout the manuscript. Thus, when describing the results, the authors should comment on changes in preference and how this can be interpreted as a change in social memory retrieval. Several critical experimental details such as the total time of presentation to the mother and non-mother stimulus mice are also lacking in the manuscript. The in vivo e-phys results are interesting as well but even more succinct with no proposed mechanism as to how abGCs could regulate SWR and PAC in CA2.

In response to these comments, we provide raw interaction times in a new Figure (Fig. S1). We also provide more information about the experiments and figures in the revision. We explain the rationale for our behavioral interpretations and discuss proposed mechanisms for how abGCs regulate SWR and PAC.

The manuscript is well-written with the appropriate references. The choice of the behavioral test is somewhat debatable, however. It is surprising that the authors chose to use a direct presentation test (presentation of the mother and non-mother in alternation) instead of the classical 3-chamber test which is particularly appropriate to investigate social preference. Since the authors focused exclusively on this preference, the 3-chamber test would have been more adequate in my opinion. It would greatly strengthen the results if the authors could repeat a key experiment from their investigation using such a test. In addition, the authors only impaired the mother's memory. An additional experiment showing that disruption of the abGCs to CA2 circuit impairs social memory retrieval would allow us to generalize the findings to social memories in general. As the manuscript stands, the authors can only conclude the importance of this circuit for the memory of the mother. Developmental memory implies the memory of familiar kin as well.

We selected the direct social interaction test because it allows for more naturalistic social behaviors than measuring investigation times toward social stimuli located inside wire mesh containers. We also decided to focus our studies on the retrieval of mother memories because these are likely the first social memories to be formed. We emphasize that our results cannot be generalized to memories of other social stimuli but given studies on recent social memory formation and retrieval in adults that manipulate abGCs and CA2 separately, we feel that it is likely that this circuit is involved in these functions as well. However, we specify throughout the manuscript that our experiments can only tell us about mother memories. We have also changed the title to reflect this.

The in vivo ephys section (Figure 3) is interesting but even more minimalistic and it is unclear how abGCs projection to CA2 can contribute to SWR and theta-gamma PAC. In Figure 1, the authors suggest that abGCs project preferentially to PV+ neurons in CA2. At a minimum, the authors should discuss how the abGCs to PV+ neurons to CA2 pyramidal neurons circuit can facilitate SWR and theta-gamma PAC.

We have divided Figure 3 into two figures (Figures 3 and 4) and revised the electrophysiology section of the results section. In the revised paper, we now discuss how abGC projections to PV+ interneurons may facilitate SWR and PAC.

Finally, proposing a function for 4-6-week-old abGCs projecting to CA2 begs two questions: What are abGCs doing once they mature further, and more generally, what is the function of the DG to CA2 projection? It would be interesting for the authors to comment on these questions in the discussion.

In response to these comments, we discuss possible answers to these interesting questions.

**Recommendations for the authors:**

**Reviewer #1:**
Specifically, in Figure 1, for the analysis of the synapses formed between abGCs and CA2PNS (as identified by PCP4 expression) and CA2 PV+ cells (as identified by cre-dependent AAV-mCherry expression) in PV-cre line. In panels c and d the soma of a CA2 PN cell is shown, as well as the soma of a PV cell is shown. Why was the soma analyzed? What relevance is there for this? It is my understanding that synapses form on dendrites- this would be much more relevant to show, in my opinion. Also, the methods for panels e and f state that the 3R-Tau+ intensity was analyzed only in stratum lucidum. (There was a normalization for the overall 3R-Tau intensity in SL of CA2 that was obtained by dividing the 3R-Tau intensity of corpus callosum). I don't understand then how a comparison of 3RTau intensity could have been done for CA2 PN soma. There are no CA2 PN soma in stratum lucidum. (This is fairly clearly shown in Figure 1aiii, with the PCP4 staining showing the soma in the somatic layer... not in stratum lucidum). What is being analyzed here?If the 3R-Tau intensity for dendrites is higher for PV cell dendrites, an example image of dendrites would be very helpful. How was the CA2 PV cell dendrite delimited from the CA2 PN dendrites at 40x magnification for the 3R-Tau intensity? Why were pre-synaptic puncta not examined? Is it possible to determine the post-synaptic target with these methods? This result could be particularly interesting, but I find it very difficult to understand the quantification or the justification behind it. To truly know if a cell is getting a connection, the best method would be to perform whole-cell patch clamp recordings of the post-synpatic target cells and use optogenetics of the abGCs. I understand that perhaps this may be beyond the scope of the paper, but it is a severe limitation for these results.

We have eliminated the cell body measures from Figure 1 and focus instead on the dendrite measures, which we agree are more relevant. We now provide high magnification example images of pyramidal cell (PCP4+) and PV+ interneuron (GFP+) dendrites in Figure 1. We thank the reviewer for pointing out the error about the stratum lucidum as some of the dendrites analyzed are located in the pyramidal cell layer. In addition, neither PCP4 nor GFP label the full extent of dendrites emanating from CA2 pyramidal cells or PV+ interneurons respectively. We mention this in the revised manuscript because abGC projections to more distal dendrites might show a different pattern than that which was observed for proximal dendrites. We also provide more details about how the dendrites were delimited for the analysis, and mention that these results cannot definitively inform us about whether functional synaptic connections have been formed.

Canulation over CA2 is potentially not specific to CA2 terminals. It would be optimal if the authors had some histology demonstrating specific cannula placement, as these surgeries are really tough to get perfectly centered over CA2. Even if it is perfectly centered, how much would the CNO diffuse into CA3? I think that given the methodology, the authors really need to consider that the behavioral results are not only a result of blocking abGC terminals in CA2 alone. Would it really change much if the abGC terminals are also silenced in CA3a/b as well? The McHugh lab has shown that area CA3 is also playing a role in social memory (Chiang, M.-C., Huang, A. J. Y., Wintzer, M. E., Ohshima, T. & McHugh, T. J. A role for CA3 in social recognition memory. Behav Brain Res 354, 2018). It may be that both areas CA2 and CA3 are important for the phenomenon being demonstrated in Figure 2. I think the impact of the study is just as interesting, as this examination of early social memories is very interesting and nicely done. In fact, areas CA2 and CA3 may be acting together (please see Stöber, T. M., Lehr, A. B., Hafting, T., Kumar, A. & Fyhn, M. Selective neuromodulation and mutual inhibition within the CA3-CA2 system can prioritize sequences for replay. Hippocampus 30, 1228-1238, 2020).

We agree that it is possible that CNO infusions targeted at the CA2 would also influence CA3a/b and have revised the paper to include this possible interpretation. We also cite the suggested paper on CA3 involvement in social memory (Chiang et al., 2018) and the paper on CA2-CA3 interactions (Stöber et al, 2020).

Figure 3 is packed with information, but not communicated in a reasonable way. Much more information and a description of the experimental protocol need to be presented.Furthermore, why are there no example traces for the SWRs recorded? There should be more analysis than just a difference score and frequency. How is j, k, and l analyzed and interpreted? Why no example traces there? Also, the n's seem way too small for Figure 3mr. Are there only 32 or three animals used for some of these conditions? This is insufficient in my opinion to conclude much for a 5-minute interaction.

In response to this concern, we have divided Figure 3 into 2 figures – Figure 3 and Figure 4. InFigure 3, we provide example traces for SWRs, with additional SWR data presented in Figures S3 and S4, including data to complement the difference score data in Figure 3. In Figure 4, we include traces of phase amplitude coupling. We also provide more information in the methods about how the phase amplitude coupling data were analyzed. For Figure 4, we used methods described by Tort et al., 2010 to produce a modulation index, which is a measure of the intensity of coupling between theta phase and gamma amplitude. This method additionally allows for visualization of how gamma amplitude is modified across individual theta phase cycles. Regarding the question about n sizes in the 10-12 week abGC group (Fig. 3), the numbers are lower than in the 4-6 week abGC group because by 6 weeks after the first set of recordings, the electrodes in some of the mice were no longer usable. The n sizes for this specific study are 4-5 per group for Nestin-cre mice; 7-8 for Nestin-cre:Gi. This is now clarified in the figure legend.

The discussion section of this paper does not put these results into a broader context with the field. There are other studies examining abGCs and their roles in novelty and memory formation (the work from Juna Song's lab, for example). These should be properly mentioned and discussed.

In response, we have added discussion on the roles of abGCs in nonsocial novelty and memory formation and have cited papers from the Song lab.

In the figure legend for Figure 2, there is no specific explanation for panel h. Perhaps the label is missing in the legend.

We thank the reviewer for noting this error and now include a description in the revised manuscript.

**Reviewer #2:**
Adding more quantifications (single cells, isolating data during interactions versus noninteraction times) would help understand the results better. In the lack of this, adding a more clear rationale (even if only through the presentation of hypotheses) in between the transitions of the different results sections would make the study easier to read.

In response to this comment, we have added transition sentences between results sections to clarify the rationale and make the manuscript easier to understand.

**Reviewer #3:**
Line 110: "Hippocampal phase-amplitude coupling (PAC) and generation of sharp waveripples (SWRs) have been linked to novel experience, memory consolidation, and retrieval (Colgin, 2015; Fernandez Ruiz et al., 2019; Meier et al., 2020; Joo and Frank, 2018; Vivekananda et al., 2021). The DG is known to influence hippocampal theta-gamma coupling and SWRs (Bott et al, 2016; Meier et al., 2020), yet no studies have examined the influence of abGCs on these oscillatory patterns." This information comes too early in the result section and is somewhat confusing.

In response to this comment, we have moved this information and provided a better description.

Line 118: "we found that mice with normal levels of abGCs can discriminate between their own mother and a novel mother." Be more descriptive of the results (present the raw interaction times with the statistical test to compare them), this is the conclusion.

In response to this comment, we provide the raw interaction times in a new Figure (Fig. S1) and describe the results in more detail.

Line 121: "These effects were not due to changes in physical activity". Be more specific. Did you subject the mice to a specific test? If not, how did you calculate locomotion? The data presented in the supplementary figure 1a only states the % locomotion.

Locomotion was manually scored whenever an animal moved in the testing apparatus. Speed was not recorded. Total locomotion was divided by trial duration to create a % locomotion measure. We have added these details to the methods.

Line 124: "Coinciding with the recovery of adult neurogenesis, GFAP-TK animals regained the ability to discriminate between their mother and a novel mother". Explain how the difference in interaction time can be interpreted as the ability to discriminate. You could also compute the discrimination index used by several other laboratories (difference of interaction normalized by the total interaction time).

In response to this comment, we describe how the difference in interaction time can be interpreted as the ability to discriminate between novel and familiar mice.

Line 133: "Targeted CNO infusion in Nestin-Cre:Gi mice enabled the inhibition of GiDREADD+ abGC axon terminals present in CA2." Provide data or references to support this claim. Injection of a dye of comparable size to CNO would help. Otherwise, mention that nearby CA3a could be affected as well.

We cannot rule out that nearby CA3a was affected by our cannula infusions of CNO into CA2. Furthermore, since dyes likely diffuse at different rates than CNO, we believe that a dye injection would not eliminate this concern completely. Therefore, we have revised the paper to acknowledge the likelihood that the CNO infusion affected parts of CA3 in addition to CA2. We also changed the title to focus more on the CA2 electrophysiological recordings, which we know were obtained only from the CA2.

Line 150: "When reintroduced to the now familiar adult mouse 6 hours later, after the effects of CNO had largely worn off". Provide data or references supporting this claim.

In response, we cite articles that show behavioral effects of CNO DREADD activation are returned to baseline 6 hrs later.

Line 165: "We found that SWR production is increased during social interaction, with more SWRs produced during novel mouse investigation, presumably during encoding social memories, than during familiar mouse investigation, presumably during retrieval of developmental social memories". How does this compare to the results in Oliva et al, Nature 2021?

The Oliva et al 2021 paper recorded CA2 SWRs during home cage and during post-social stimulus exposure periods of sleep. The timing of the study does not coincide with the measures we made, but we cite the paper.

Line 168: "Inhibition of abGCs in the presence of a social stimulus". How does silencing abGC impact CA2 pyramidal neurons' firing rate?

The direct answer to this question is unknown because we did not measure single units, but based on studies done in the CA3, it is likely that firing rate in CA2 would increase.

Line 203: "abGCs possess a time-sensitive ability to support retrieval of developmental social memories." Can you speculate on the function of the cells later on?

In the revised paper, we speculate about the function of abGCs after they mature and no longer support retrieval of developmental social memories.

Line 229: "GFAP-TK mice were group housed by genotype". Why not housed them with CD1 littermates?

We housed these mice according to genotype to avoid having mice with different levels of abGCs (GFAP-TK + VGCV and CD1 + VGCV) living together in social groups. We did this to avoid potential differences that might emerge in social behavior.

Line 237: "Adult TK, Nestin-cre, and Nestin-cre:Gi offspring underwent a social interaction test in which they directly interacted with the mother". Specify how long was the social interaction time.

In the revised manuscript, we specify that mice interacted with each social stimulus for 5 minutes.

Line 240: "After a 1-hour delay spent in the home cage". Were the mice single-housed or with their littermates during this delay?

In the revised manuscript, we indicate that mice were put back into the home cage with their cagemates during the 1 hr delay period.

Line 241: "The order of stimulus exposure was counterbalanced in all tests." Can you show some data to confirm that the order of presentation did not impair the interaction? Have you considered using your own version of the classical 3-chamber test in order to assess directly the preference for one or the other female mouse?

Our data suggest that the order of testing is not responsible for the observed results. Across all experimental groups without an abGC manipulation (i.e., all direct social interaction assays excluding VGCV+ GFAP-TK trials and CNO+ Nestin-cre:Gi trials), ~84.4% of animals demonstrate a social preference for the novel mother over the mother (CD1 + GFAP-TK VGCV- cohort: 28/33; CD1 VGCV+ cohort: 17/17; CD1 and TK recovery cohort: 24/31; Nestin-cre and Nestin-cre:GI 4-6-week-old abGC cohort: 77/95; 10-12-week-old abGC cohort: 49/55; Total = 195/231 mice with an investigation preference for the novel mother). If stimulus presentation order were to bias social investigation preference toward the first stimulus presented, we would expect the percentage of animals demonstrating a social preference for each stimulus to be around 50%, as roughly half the animals were first exposed to the mother with the other half first exposed to the novel mother. The social novelty preference percentage reported above is comparable to percentages we observe in our lab's novel to familiar social interaction experiments, in which all animals are first exposed to a novel conspecific. We have yet to conduct experiments testing adults using the modified 3-chamber assay described in Laham et al., 2021.

Statistics: The statistical tests used throughout the paper are appropriate but their description is too cursory. Please provide F values and specify the name of the tests used in the figure legends before giving the exact p values.